# CFD-Based Study on the Airflow Field in the Crushing Chamber of 9FF Square Bale Corn Stalk Pulverizer

**Jie Zhang** [1,†]**, Xiang Tian** [1,†]**, Chao Zhao** [1]**, Xiuzhen Yu** [1]**, Shiguan An** [1]**, Rui Guo** [2,*] **and Bin Feng** [1,*]

1   Institute of Agricultural Mechanization, Xinjiang Academy of Agricultural Sciences, Urumqi 830091, China
2   Route Department, Xinjiang Division, China Southern Airlines Technical Branch, Urumqi 830002, China
*   Correspondence: goodrui2023@163.com (R.G.); fbxjnky@stu.nwupl.edu.cn (B.F.)
†   These authors contributed equally to this work.

**Abstract:** During the steady operation of the 9FF square bale corn stalk pulverizer, the rapidly rotating spindle drives the hammers and impellers to form a complicated airflow field environment in the crushing chamber. The flow field characteristics in the crushing chamber can affect the motion law of stalks, thus influencing the pulverizing effect of the hammer blades on stalks and the ejection of materials. Based on establishing the Computational Fluid Dynamics (CFD) calculation model of the crushing chamber in the 9FF square bale corn stalk pulverizer, in this paper, the effect of three groups of components, such as the hammer rack and blades, hammer rack and blades and the sieve, as well and the sieve and the impeller, on the distribution characteristics of airflow field in the crushing chamber. The simulation results show that the hammer piece group, screen, and impeller in the crushing room can effectively improve the crushing quality of materials, with the crushing efficiency and conveying efficiency at the highest speed of 3000 r/min.

**Keywords:** square bale corn stalks; pulverizer; airflow field; CFD

## 1. Introduction

Corn stalk is the main source of coarse fodder for cattle and sheep breeding in Xinjiang, China [1–3]. With the improvement of farmland mechanization and the widespread application of square bale mechanization technology [4], currently, the corn straw used for feed in cattle and sheep breeding, whether it is household-type free-range breeding or large-scale shed-feeding breeding, is basically stored in square bales [5]. In view of the problem that the energy consumption, output, material length after crushing, and other aspects of the existing pulverizers in China cannot well meet the requirements for corn stalk crushing for square bales [6], the authors developed and designed a 9FF square bale corn stalk pulverizer suitable for direct crushing of corn stalks to make square bales.

## 2. Analysis of the Pneumatic Conveying of Corn Stalks in the Crushing Chamber

### 2.1. Working Principle

The simplified interior structure and the whole machine of the crushing chamber of the 9FF square bale corn stalk pulverizer are shown in Figure 1. During the material feeding process, the materials come into contact with the rapidly rotating hammers and are drawn into the crushing chamber through friction at the top of the hammer blades. Due to the significant speed difference between the materials and the hammer blades, the materials fed obliquely at an angle of 15° from the horizontal direction are stricken into short segments by the hammer blades and restricted by the sieve, they are continuously hammered and sheared by the hammer blade group, and finally, the crushed materials meet the quality standards are thrown and sucked to the wall surface of the crushing chamber under the action of centrifugal inertia and negative pressure drag force. Then, the stalk materials are crushed into fine filaments or short stalks under the rubbing of the hammer blade group

and the sieve, and under the combined action of the thrust of the rapidly rotating impeller and radial airflow, they are finally thrown to the outlet hole.

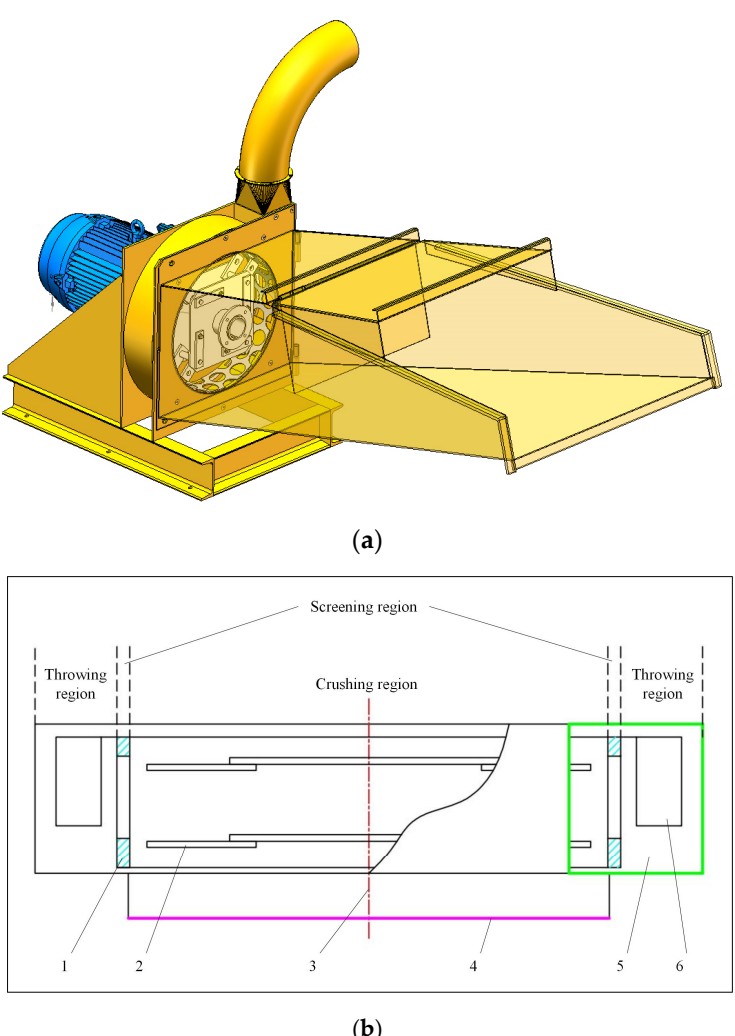

(**a**)

(**b**)

**Figure 1.** Simplified interior structure and the whole machine of the pulverizer. (**a**) Three-dimensional modeling diagram of the pulverizer. (**b**) Interior structure of the crushing chamber. Note: 1. Sieve; 2. Hammer blade group; 3. Rotating spindle; 4. Feeding inlet; 5. Outlet; 6. Throwing blade.

## *2.2. Analysis of Pneumatic Conveying*

The hammer blades rotate at high speed in the crushing chamber and form a circulation layer. After sufficient shearing and rubbing, the feeding stalks are crushed into light and short stalks or thin strips, and under the effect of the high-speed airflow in the circulation layer, the materials are suspended [7–9]. As shown in Figure 2, the corn stalks were divided into two regions for analysis on pneumatic conveying. In the radial plane, region I is the crushing region, where the materials were subjected to the tangential airflow speed $V_{\tau 1}$ and radial airflow speed $V_{\sigma 1}$, whose sum speed is $V_{\alpha 1}$. The hammers' striking speed on corn stalks was $V_h$, which is in the same horizontal plane as the sum speed $V_{\alpha 1}$. Region II is the throwing region, where the materials were subjected to the tangential airflow speed $V_{\tau 2}$ and radial airflow speed $V_{\sigma 2}$, whose sum speed is $V_{\alpha 2}$. The throwing speed from the throwing blades is $V_p$. The effect of gravity in both regions was ignored, and the airflow forces in the radial plane on the materials are $F_{p1}$ and $F_{p2}$ [10]:

$$F_{P1} = C_0 A \frac{K_a (V_h - V_{\alpha 1})^2}{2g} \tag{1}$$

$$F_{P2} = C_0 A \frac{K_a (V_P - V_{\alpha 2})^2}{2g} \tag{2}$$

where $C_0$ is the material resistance coefficient; $A$ is the projected area of materials in the motion direction, m$^2$; $K_a$ is the density of the air in the crushing chamber, Kg/m$^3$; $g$ is the acceleration of gravity, m/s$^2$.

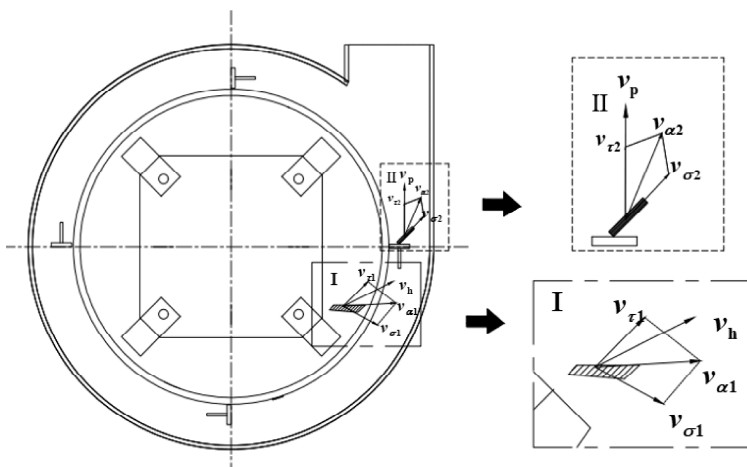

**Figure 2.** Force analysis of materials' airflow field.

It can be obtained from Equations (1) and (2) that, in the radial plane of the crushing chamber, the materials were affected by material resistance coefficient, projected area, material motion speed difference and airflow speed difference in the flow field. Among them, stalk materials were affected by material resistance coefficient and projected area; the material motion speed difference and airflow speed difference were affected by air density, airflow speed, straight-line distance between material and center of rotation, as well as motion speed at difference rotating speeds of hammer blades and throwing blades in regions I and II.

Meanwhile, since the axial length of the crushing chamber is far less than its radial length, there is mainly the action of radial airflow within the crushing chamber. In the axial plane, axial conveying mainly relies on the negative pressure suction at the center of rotation produced by the high-speed rotation of components in the crushing chamber, as well as the force of gravity on the materials, to realize the continuous feeding of materials into the crushing chamber through the inlet.

The crushing region and sieving region are constituted by the hammer rack and blades, and the sieve can increase the effective contact time between the hammer blade group and materials, thus enhancing the rubbing and crushing effect. The throwing region constituted by the throwing blades has increased the radial speed gradient of the crushing chamber, which is conducive to material conveying, thus avoiding clogging and enhancing productivity.

### 3. Introduction to the Computational Fluid Mechanics

*3.1. Basic Governing Equation in the Flow Field Simulation Analysis*

In the process of numerical simulation of flow fields, the three governing equations of fluid flow are usually expressed in the form of continuity equations, momentum equations, and energy equations, which are also the theoretical basis for simulation in this study [11].

In analyzing the internal flow field of the crushing chamber of the square bale corn stalk pulverizer, the internal flow field is generally taken as an incompressible turbulent flow [12]. Regardless of temperature changes and gas compression, only the constraints of the governing equations of mass and momentum conservation are considered.

Continuity equation:

All problems associated with flow should follow the law of mass conservation. That is, the mass of fluid flowing into a closed area per unit of time is equal to the net mass flowing out of the area. The continuity equation for fluid flow is as follows:

$$\frac{\partial \rho}{\partial x} + \frac{\partial(\rho u)}{\partial x} + \frac{\partial(\rho v)}{\partial y} + \frac{\partial(\rho w)}{\partial z} = 0 \tag{3}$$

where $\rho$ is the density of the fluid, Kg/m$^3$; t is the time, s; $u$ is the component of velocity in the $x$-direction of a rectangular coordinate system, m/s. $v$ is the component of the flow velocity in the $y$-direction of the rectangular coordinate system, m/s; $w$ is the component of the flow speed in the $z$-direction of the rectangular coordinate system, m/s;

Momentum equation:

All problems associated with the flow should follow the law of conservation of momentum, which means that the rate of change of the momentum of the fluid in the control body over time is equal to the sum of various external forces acting on the fluid control volume. The expressions of the momentum conservation equation in the $x$-, $y$-, and $z$-directions are the following:

$$-\frac{\partial \rho}{\partial x} + \frac{\partial \tau_{xx}}{\partial x} + \frac{\partial \tau_{yx}}{\partial y} + \frac{\partial \tau_{zx}}{\partial z} + F_x = \frac{\partial(\rho u)}{\partial t} + div(\rho u U) \tag{4}$$

$$-\frac{\partial \rho}{\partial y} + \frac{\partial \tau_{xy}}{\partial x} + \frac{\partial \tau_{yy}}{\partial y} + \frac{\partial \tau_{zy}}{\partial z} + F_y = \frac{\partial(\rho v)}{\partial t} + div(\rho v U) \tag{5}$$

$$-\frac{\partial \rho}{\partial z} + \frac{\partial \tau_{xz}}{\partial x} + \frac{\partial \tau_{yz}}{\partial y} + \frac{\partial \tau_{zz}}{\partial z} + F_z = \frac{\partial(\rho w)}{\partial t} + div(\rho w U) \tag{6}$$

where $\rho$ is the pressure on the fluid control volume, MPa; $\tau_{xx}$, $\tau_{yx}$, and $\tau_{zx}$ are components of viscous stress in the $x$-direction, MPa; $\tau_{xy}$, $\tau_{yy}$, and $\tau_{zy}$ are components of viscous stress in the $y$-direction, MPa; $\tau_{xz}$, $\tau_{yz}$, and $\tau_{zz}$ are components of viscous stress in the $z$-direction, MPa; $F_x$, $F_y$, and $F_z$ are components of the external force on the micro-unit in the coordinate direction, MPa; $U$ is the velocity vector; $t$ is the time, s. $u$, $v$, and $w$ are components of U in the three coordinate directions.

### 3.2. Numerical Discretization Method in the Flow Field Simulation Analysis

Based on different discretization principles, CFD can be mainly divided into three parts: finite difference method (FDM), finite element method (FEM), and finite volume method (FVM). Among them, FVM divides the flow field space into many small space units, each unit acting as the finite control unit, and the derived basic equation can express the mechanical conservation relation of a single control unit. The coefficients in the equation have definite physical meanings. This solution is closer to the motion law of mechanical continuous media than the weighted residual method used in the finite element method. The discretization in finite volume method was adopted in the numerical simulation of the interior flow field of the crushing chamber of the 9FF square bale corn stalk pulverizer.

### 3.3. Introduction to the Flow Field Simulation Analysis Software Fluent

In recent decades, commercial CFD software has flourished, and Fluent has developed into one of the most comprehensive software packages in the field of CFD simulation in the world. Its extensive physical models enable users to quickly and accurately obtain CFD analysis results [13,14].

## 4. Establishment of the Model of the Internal Airflow Field of the 9FF Square Bale Corn Stalk Pulverizer

### 4.1. Establishment of the Flow Field Regions Based on Design Modeler

The internal space of the crushing chamber was taken as the calculation area for simulation. The internal space of the crushing chamber includes the hammer rack and

hammer blades, the sieve, and the impeller. The hammer blades and the impeller rotate rapidly around the axis and form a sieve structure on the wall of the crushing chamber. In order to explore the effects of working components of the crushing chamber on the whole flow field, the influence characteristics of each component on the airflow field were analyzed in this study.

SolidWorks2018 was used to simplify some models of the blade rack, and then the assembly of the hammer rack and blades, hammer rack and blades and the sieve, the impeller and the sieve were named Model 1, Model 2, and Model 3, respectively, as shown in Figure 3b–d. Then, the files were imported into the ANSYS Workbench 2020 according to the numerical simulation objectives, and the flow field regions of corresponding models were established through the software Design Modeler.

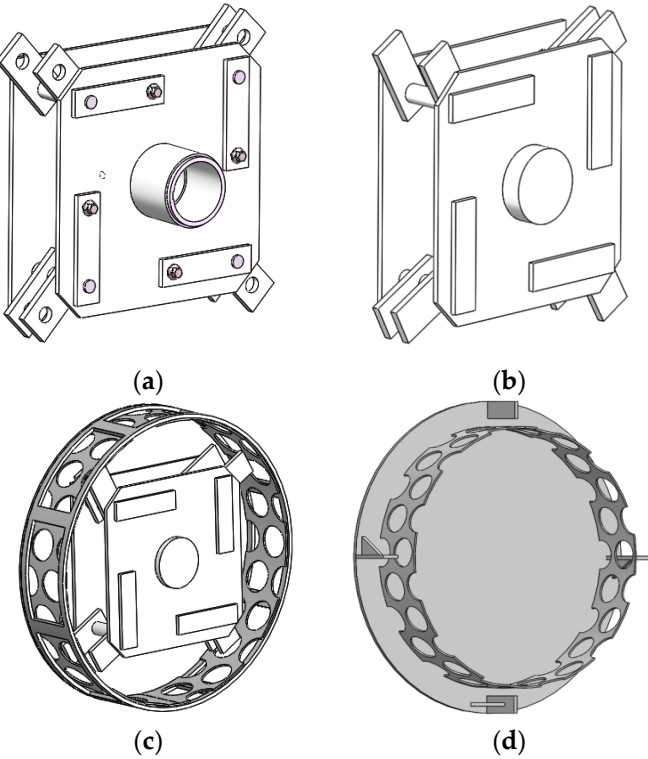

| (a) | (b) |
| (c) | (d) |

**Figure 3.** Simplified 3D models. (**a**) A 3D model of the hammer rack and blades. (**b**) A 3D model of the simplified hammer rack and blades (Model 1). (**c**) Hammer rack and blades and the sieve (Model 2). (**d**) The impeller and the sieve (Model 3).

In establishing 3D models, the internal space of the crushing chamber was simplified as follows:

1. The dimensions of the outlet and inlet were simplified, and only the key components in the crushing chamber were retained. Because the dimensions of the feeding inlet and outlet were large, in establishing the flow field regions, more calculation power would be required; thus, its dimensions were simplified to speed up the analysis speed of meshing subsequently [15].
2. Regarding the crushing chamber as a completely airtight space, all small gaps and chamfers were omitted to simplify the model structure. Due to mechanical connection, there would be slits and chamfers, which had minimal impact on the negative pressure characteristics inside the crushing chamber, so they are omitted.
3. The spindle and the hammer rack were fixed, and the hammer blades and hammer rack were regarded as an integrated whole, and the mechanical connection between all components was ignored. During the high-speed rotation of the pulverizer, due to the centrifugal effect generated by the rotation, the hammer blades would extend

along the radial direction. Because the hammer blades were not fixed on the hammer rack, the movement of the hammer blades was hysteretic. The pretest results show that the hammer blades were 5° hysteretic relative to the motion of the hammer rack, and this location relation was considered to set the location of the hammer blade group in simulation, and the hinge joint between the hammer blade group and the hammer rack is ignored. Meanwhile, the slits made by the key joint between the hammer blade rack and the hinge joint between the hammer blade group and the hammer rack would not affect the negative pressure of the flow field; therefore, the axis, hammer rack, and blades could be regarded as an integrated whole. The map of the flow field model of the crushing chamber established by the Design Modeler is shown in Figure 4.

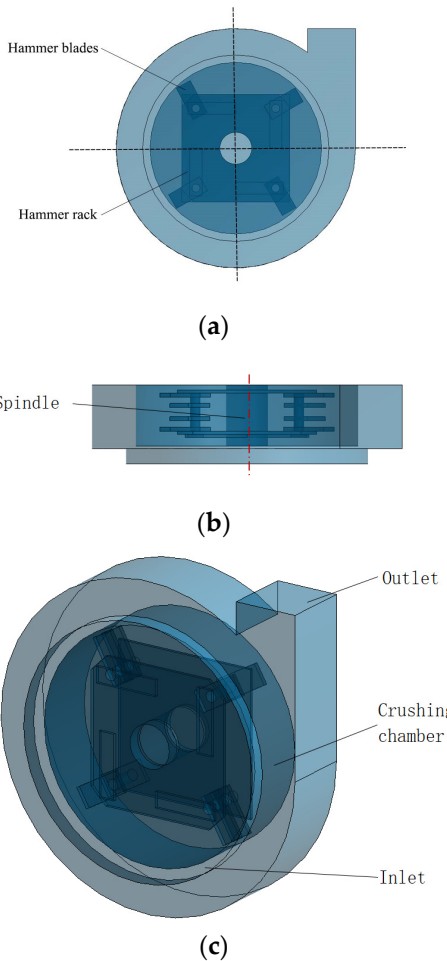

**Figure 4.** The 3D models of the flow field in the crushing chamber. (**a**) Front view of the flow field. (**b**) Arrangement pattern of the blades. (**c**) Axonometric drawing of 3D model of the flow field in the crushing chamber.

*4.2. Mesh Generation Based on Mesh*

After establishing the flow field model of the crushing chamber of the 9FF square bale corn stalk pulverizer, the model was imported into the pre-processing software Fluent and Mesh for mesh generation on the three groups of models.

(1) Establishment of the mesh model.

In establishing a model, the flow field model was divided into four regions using faces and volumes: the interior region of the crushing chamber, the hammer blade rack, the region for hammer group and impeller rotation, the inlet, and the outlet. These four regions constitute the rotation region and the static region. The rotation of area 1 wrapping the

hammer rack and blades and the rotation of area 2 wrapping the impeller were established by the Design Modeler. The rotation of areas 1 and 2 are both cylindrical and share the same rotation axis. The remaining areas, including the peripheral areas of the rotating area inside the crushing chamber, the sieve area, and the feed inlet and outlet, are all static areas.

(2)  Meshing.

Based on the demand of analysis, the adaptive function was used for unstructured meshing [16,17]. For the edges and surfaces of the hammer blade group, the hammer group and the sieve, and the impeller that contacts the flow field, as well as the rotating areas 1 and 2, the mesh control method was used on the local line, surface, and volume sizes to encrypt the meshes. For the remaining parts, due to the low demand for computational accuracy, the structure is relatively simple, and the mesh generation is relatively rough. After mesh generation by Mesh, the mesh model data are shown in Table 1, and the mesh generation results are shown in Figure 5.

**Table 1.** Mesh generation data of three groups of models.

| Model Name | Model 1: Hammer Rack and Blades | Model 2: Hammer Rack and Blades and the Sieve | Model 3: Impeller and the Sieve |
|---|---|---|---|
| Total units | 1,249,156 | 2,113,177 | 1,881,289 |
| Total nodes | 765,713 | 409,185 | 345,986 |

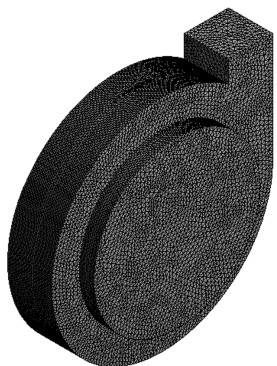

**Figure 5.** Results of mesh generation.

(3)  Mesh quality inspection.

After meshing, the quality of the meshes was inspected. Mesh quality can be inspected through the Minimum Value and Orthogonal Value. After mesh generation, the Minimum Values of the models were all above 0, showing good mesh quality. Through orthogonal quality inspection, the orthogonal quality of the three groups of models was 0.77665, 0.83705, and 0.86159, showing good mesh quality [18].

## 5. Characteristic Analysis of the Flow Field of the Crushing Chamber of the Square Bale Corn Stalk Pulverizer Based on Fluent

In this section, the CFD analysis software Fluent 19.0 was used to conduct numerical simulation research on the negative pressure characteristics and velocity characteristics of the hammer frame, hammer frame, hammer group, and screen, as well as the screen and impeller in the single-phase gas flow field in the crushing chamber of the baled corn straw pulverizer, providing a theoretical basis for the research on the flow field and structural improvement of the pulverizer.

### 5.1. Setting of Boundary Conditions

The boundary types of the mesh model were specified through the software Fluent. The specific boundary types are shown in Table 2.

**Table 2.** Boundary types of the models.

| Name of Boundary | Types of Boundary |
|---|---|
| Inlet | Pressure inlet |
| Outlet | Pressure outlet |
| Hammer rack and blades | Wall |
| The impeller | Wall |
| Sieve | Wall |
| Inner contact surface | Interface |
| Outer contact surface | Interface |

Because the hammer rack and hammer blades in the crushing chamber are rotary components, and others belong to a static area, therefore, the method of the Moving Reference Frame [19–21] was adopted to find the solution and specify the type of continuous medium on the mesh models, which are rotary region and static region, and the medium type is Fluid.

The rotary region and the static region were connected through the inner and outer contact surfaces. For the Moving Reference Frame (MRF) method, Fluent can automatically generate contact surfaces in which the inner contact surface is the rotary part, and the outer contact surface is the static part.

*5.2. Turbulence Model in the Flow Field Simulation Analysis*

(1)     Determination of flow status.

During the operation of the pulverizer, due to the irregularity of flow pass in the internal flow field, rotation of the hammer blades and the impeller, and the difference of each boundary, the trajectory of airflow cannot form regular and smooth curves; therefore, it can be determined that the flow status is turbulent flow.

(2)     Governing equation of the turbulence model.

Taking the computational model of the inner flow field as an incompressible turbulence model, in this paper, the standard k-epsilon model was adopted for the numerical simulation of the inner flow field of the crushing chamber. The standard k-epsilon model is currently the most widely used turbulence model, and it contains two basic equations, which are the equation of turbulent kinetic energy k and the equation of dissipation rate [22,23].

(3)     Parameter settings of the software Fluent.

Rotating speed of the pulverizer (rotating speed of the rotation domain): 2000 r/min, 2500 r/min, and 3000 r/min.

Acceleration of gravity: z-axis direction of the space coordinate system of the model, −9.81 m/s.

Materials: the fluid is air, and the wall is steel.

Cell zone condition: set the rotation area as the Moving Reference Frame, and the rotation axis is set to the straight line taken by (0, 0, 0) and (1, 0, 0).

Pressure-speed coupling method: SIMPLEC, other parameters are set by default;

Relaxing factors: the selection of relaxing factors is shown in Table 3 as follows.

**Table 3.** Parameter setting for relaxing factors.

| Pressure | Density | Body Forces | Momentum | Turbulent Kinetic Energy | Turbulent Dissipation Rate | Turbulent Viscosity |
|---|---|---|---|---|---|---|
| 0.3 | 1 | 1 | 0.7 | 0.8 | 0.8 | 1 |

*5.3. Results and Analysis*

Based on the study, on the inner flow field of the crushing chamber at three levels of rotating speed of the spindle, 2000 r/min, 2500 r/min, and 3000 r/min, the pressure and speed distribution of each model in the crushing chamber can be obtained to analyze the effect of rotating speed of the spindle on the flow field of the crushing area.

In order to express the key numerical results of the airflow status in the crushing chamber, three sections, S1 (Y = 0), S2 (Z = 33.5 mm), and S3 (Z = −15.5 mm) were made, and their coordinates are as follows:

S1: (71.5, 300,0), (0, 300, 0), (−52.5, 0, 0);

S2: (33.5, 300, 0), (33.5, −300, 0), (33.5, 300, 300);

S3: (−15.5, 300, 0), (−15.5, −300, 0), (−15.5, 300, 300);

where S1 is the cross section that passes through the rotating axis and is perpendicular to the direction of gravity in the crushing chamber; S2 and S3 are the longitudinal sections that are perpendicular to the rotating axis, parallel to the rear wall of the crushing chamber, and coincide with the midplane of the hammer blades. The three sections are important positions in the crushing chamber of the pulverizer, which can fully demonstrate the distribution of pressure and speed.

1.  Effect of hammer rack and blades on negative pressure and airflow speed in the crushing chamber.

    (1)  Negative pressure characteristics in the crushing chamber.

When the rotating speed of the pulverizer was 2000 r/min, 2500 r/min, and 3000 r/min, the effect of the hammer rack and blades on the negative pressure characteristics in the crushing chamber can be expressed by the negative pressure characteristics of sections S1–S3, as shown in Figure 6.

It can be obtained by analyzing the pressure nephogram of the hammer rack and blades at different levels of rotating speed in the crushing chamber of the pulverizer that the following is true.

During the operation of the pulverizer, the rapidly rotating hammer rack and blades had significant disturbance to the airflow in the pulverizer, and the maximum value of the negative pressure increased with the increase of the rotating speed of the spindle. The value of negative pressure decreased along the radial direction of the rotation axis to the wall surface, while in the axial direction of the rotation axis, there was little change in negative pressure value.

Negative pressure is mainly concentrated around the rotation axis and hammer rack. Additionally, there is a small area of negative pressure around the autorotation pin shaft of the hammer blades, where the maximum negative pressure value is mainly concentrated. The reason is that there is a distance between the autorotation pin shaft and the center of rotation, and the high linear speed would produce higher negative pressure to reverse the rotation of the pin shaft after the airflow rotates around the autorotation pin shaft. Moreover, the negative pressure is higher closer to the surface of the autorotation pin shaft. When it is 0.5D over the autorotation pin shaft, the rotation reverses, the negative pressure changes significantly, and the negative pressure value decreases.

For every 500 r/min increase in the rotating speed of the spindle, the negative pressure value in the crushing chamber increased by about 0.8 KPa.

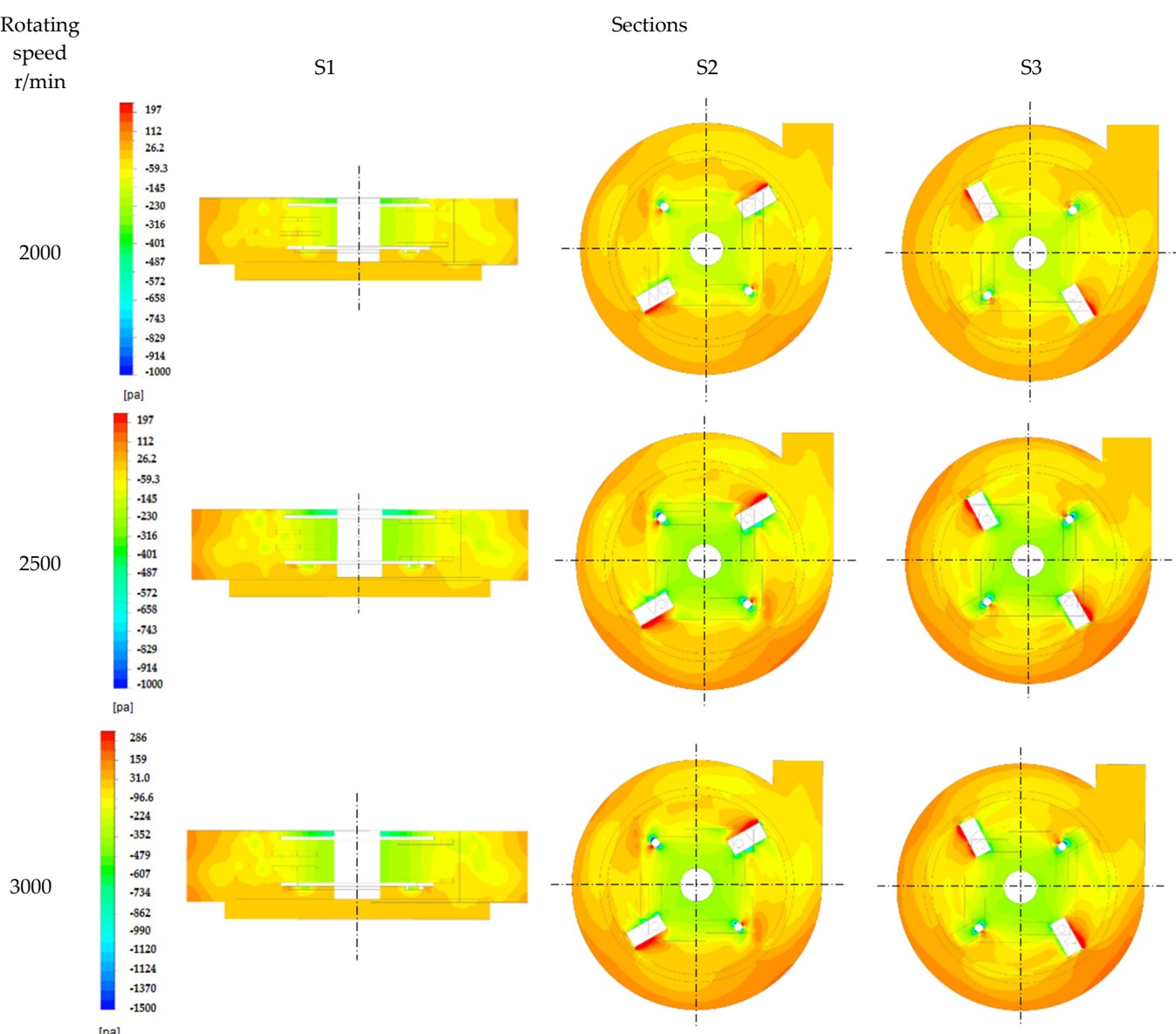

**Figure 6.** Pressure nephogram of sections S1–S3 of Model 1 at different rotating speeds.

(2)   Characteristics of airflow speed in the crushing chamber.

When the rotating speed of the pulverizer was 2000 r/min, 2500 r/min, and 3000 r/min, the effect of the hammer rack and blades on the airflow speed characteristics in the crushing chamber can be expressed by the airflow speed characteristics of sections S1–S3, as shown in Figure 7.

It can be obtained by analyzing the speed vector nephogram of sections S1–S3 of the hammer rack and blades at different levels of rotating speed in the crushing chamber that the following is true.

The airflow speed in the crushing chamber increased due to the increase in spindle speed. In the peripheral edge area of the hammer group, the airflow velocity was higher than that of the other parts, while in the axial direction of the rotation axis, the airflow velocity value had little change. Due to the counterclockwise rotation of the hammer rack and hammer blades, there is significant air disturbance in the flow field at the shortest linear distance between the inlet and the outlet, resulting in an increase in airflow velocity in this area and the maximum airflow velocity also occurred in this area. For every 500 r/min increase in spindle speed, the flow field speed value increased by about 14 m/s.

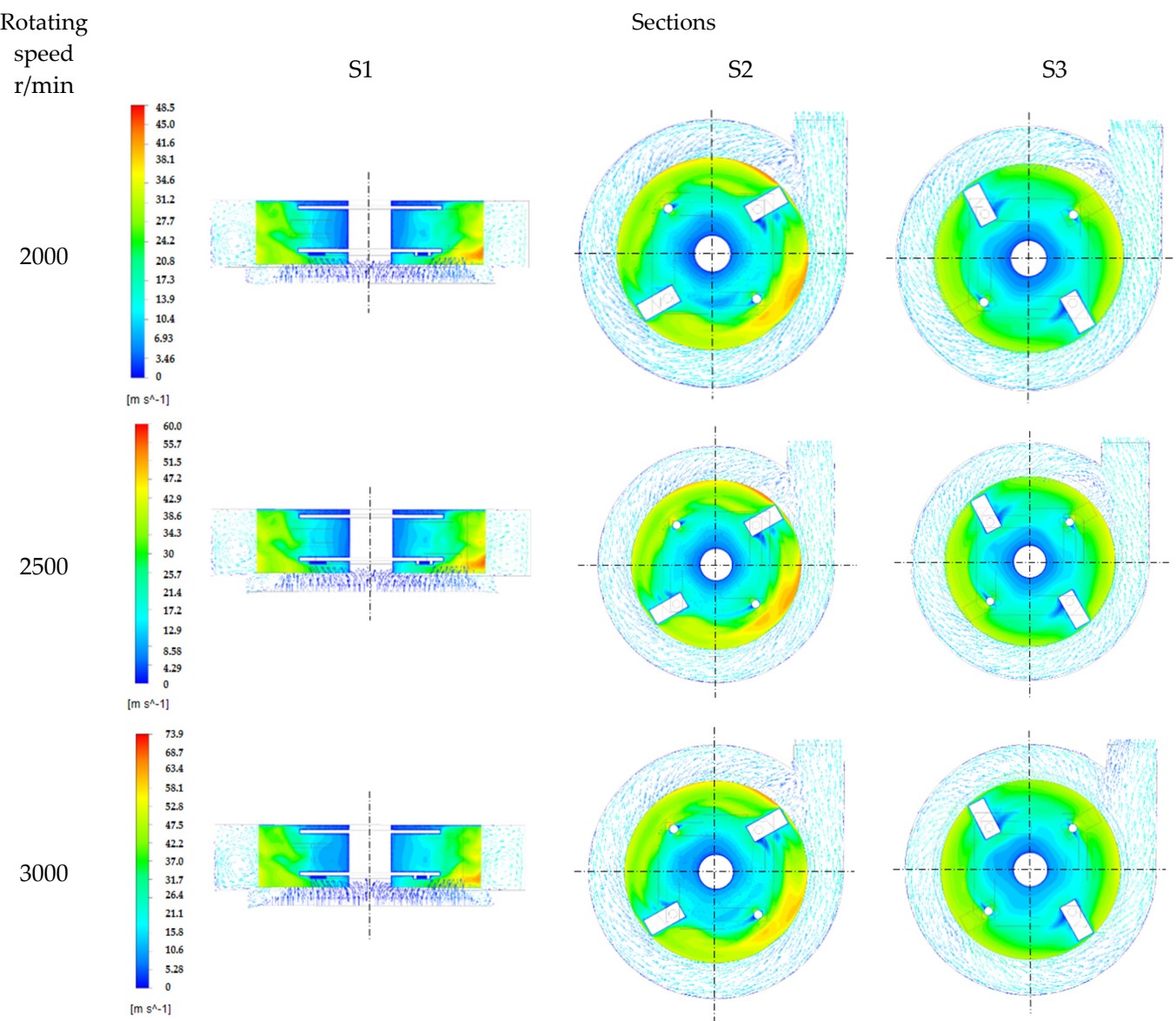

**Figure 7.** Speed nephogram of sections S1–S3 of Model 1.

Due to the great distance between the outer edge of the hammer blade group and the wall of the crushing chamber, the hammer rack and blades could crush the materials after they reached the crushing chamber. However, the probability of contact between the hammer blades and the materials was low, making it difficult to ensure the crushing quality. At three different speeds, the average airflow speed at the outlet was 13.35 m/s, 17.15 m/s, and 21.10 m/s. As the speed increased, the airflow speed at the outlet also increased. With each increase of 500 r/min, the average airflow speed at the outlet increased by about 4 m/s, which had a poorer conveying ability on the materials. After crushing, the materials were prone to accumulate and block inside the crushing chamber, resulting in lower productivity.

2. The effect of hammer rack and blades on negative pressure and speed characteristics in the crushing chamber.

(1) Negative pressure characteristics in the crushing chamber.

By analyzing the pressure nephogram of sections S1–S3 of Model 2 at different levels of rotating speeds in the crushing chamber of the pulverizer, as shown in Figure 8, the following conclusions can be drawn.

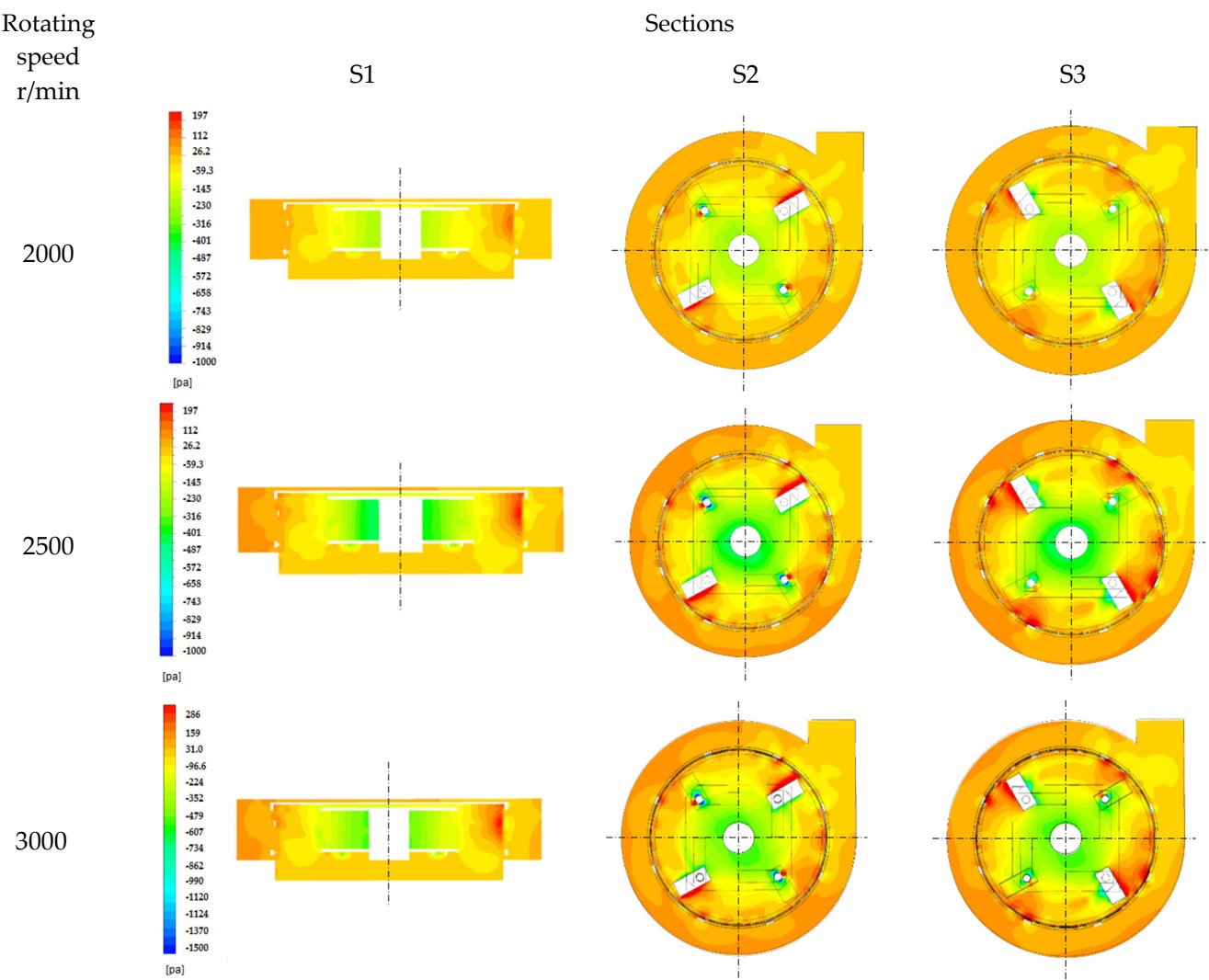

**Figure 8.** Pressure nephogram of sections S1–S3 of Model 2.

Model 2 is fixed with a sieve compared with Model 1. Although the maximum negative pressure in the crushing chamber increased with the increase of the spindle speed of the crusher, the maximum negative pressure values in the crushing chamber at different speeds decreased compared with that of Model 1. The reason is that the sieve wall had a blocking effect on the radial flow field of the rotating axis.

In the anticlockwise rotation of the hammer blades, the positive pressure value on the windward side is high. With the increase in rotating speed, the positive pressure value also increased continuously, forming a high differential pressure before and after the hammer blades, which could destroy the formation of the material circulation layer in the sieve and improve the material crushing quality and efficiency in the crushing chamber. Meanwhile, the positive pressure airflow acted on the wall surface of the sieve in the direction of hammer movement, causing a temporary increase in the positive pressure value on some sieve meshes. The airflow pressure value after going through the sieve decreased but kept positive, which could improve the screening efficiency of crushed stalks and reduce power consumption.

(2) Characteristics of the speed characteristics in the crushing chamber.

It can be obtained by analyzing the speed vector nephogram of sections S1–S3 of the hammer rack and blades and the sieve at different levels of rotating speeds in the crushing chamber that the following is true.

Due to the addition of the sieve, the maximum values of rotating speed all reduced slightly, with the highest speed reduction not exceeding 7.5%. Among them, the maximum speed reduction was at 3000 r/min, which is due to the simplification of the boundary conditions of the airflow field at the inlet and outlet of the crushing chamber. The increase in rotating speed increased the negative pressure in the central area of the rotation axis, causing flowback at the outlet and eddy currents in the intersection area of the crushing chamber and the outlet, causing a slight reduction of the maximum airflow speed.

With the increase in the rotating speed, the range of airflow speed in the rotating area of the hammer blades increased from 20.81 to 34.74 m/s to 35.54 to 51.02 m/s. The kinetic energy of the hammer blade group was higher, which is more conducive to the full crushing of stalk materials in the crushing chamber and improved the crushing quality and efficiency of the pulverizer.

From the speed fields of different cross sections in Figure 9, it can be seen that the flow speed gradient from the wall of the crushing chamber towards the rotor's axis was obvious, and the sieve wall had a significant blocking effect on the airflow speed. The airflow speed after going through the meshes was relatively high, which is conducive to getting qualified crushed stalks. However, the airflow speed from the sieve to the wall of the crushing chamber was relatively low, and the airflow after screening was turbulent airflow, which was extremely unstable and led to energy dissipation, resulting in a significant decrease in the average airflow speed at the outlet, which was 8.92 m/s, 11.63 m/s, and 14.74 m/s. This is not conducive to the conveying and timely discharge of stalks and would easily cause material congestion and accumulation, reducing productivity.

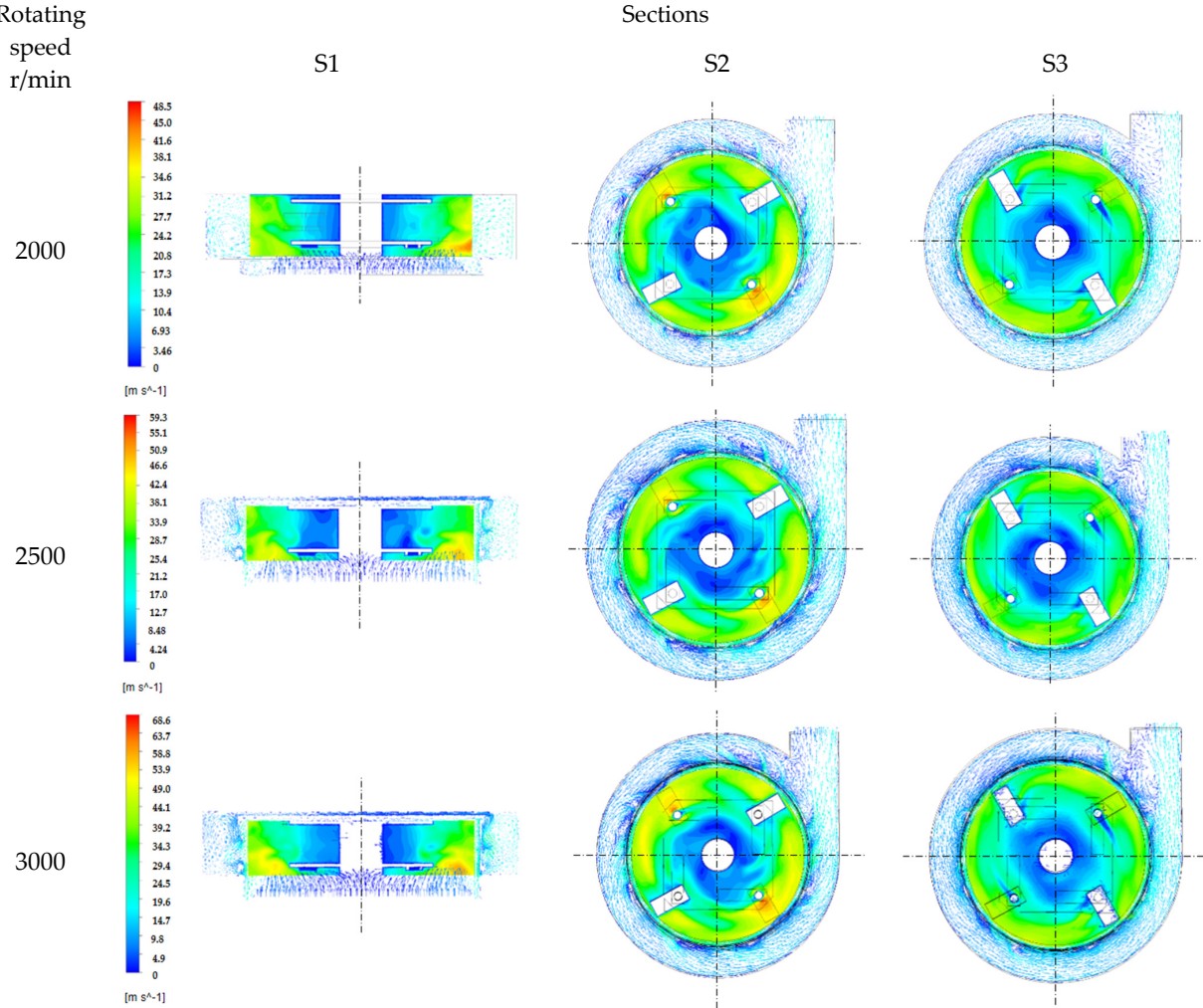

**Figure 9.** Speed nephogram of sections S1–S3 of Model 2.

3.  The effect of the impeller and the sieve on negative pressure and speed characteristics in the crushing chamber.

    (1)  Negative pressure characteristics in the crushing chamber.

    It can be obtained through analyzing the pressure nephogram of section S2 of Model 3 at different levels of rotating speeds in the crushing chamber of the pulverizer that the following is true.

    An impeller was fixed in the crushing chamber. During the rapid rotation of the impeller, in the direction from the chamber wall to the rotation axis, there was significant pressure descending. Taking the sieve as the boundary, the area from the sieve to the rotation axis was the negative pressure area, the area from the sieve to the chamber wall was the alternating area of positive and negative pressures, and the positive pressure was the highest at the windward side and had a strong conveying effect on the materials after screening. The leeward side near the wall surface of the sieve had the highest negative pressure, which has strong adsorption to the materials in the crushing chamber and could guide the qualified materials to go through the sieve. The higher the rotating speed is, the stronger the adsorption of the negative pressure.

    With the increase of the rotating speed of the spindle, the negative pressure generated by the rotation of the impeller increased accordingly. When the impeller blades rotated to the position adjacent to the outlet and the rotation area, it was easy to generate backflow there, and the local negative pressure value at the outlet increased. The reason is the simplification of the outlet before simulation, in which the outlet was shortened to only one-tenth of the actual length and was taken as the boundary condition for the pressure at the outlet. As a result, there was negative pressure generated by the backflow near the outlet. When the outlet was extended to half of the actual length, the problem of reflux was resolved.

    (2)  Speed characteristics in the crushing chamber.

    According to the analysis and study of the velocity vector nephogram of section S2 of Model 3 at different speeds in the crushing chamber, it can be seen that after the impeller was set in the flow field between the sieve and the wall of the crushing chamber, the throwing blades installed on the impeller rotated at high speed, driving the flow field in this area to rotate rapidly. Compared with Model 1 and Model 2, the airflow speed in this area was significantly increased, and the airflow speed at the outlet was also significantly increased. It can be seen from the sections A-A, B-B, and C-C in Figures 5–11 that the maximum airflow speeds at the outlet at three different rotating speeds were 38.79 m/s, 49.53 m/s, and 61.60 m/s, which is conducive to the conveying of crushed materials and avoids the congestion of materials in the area after screening. At the same time, the high-speed airflow generated by the impeller could also timely throw materials to the outlet.

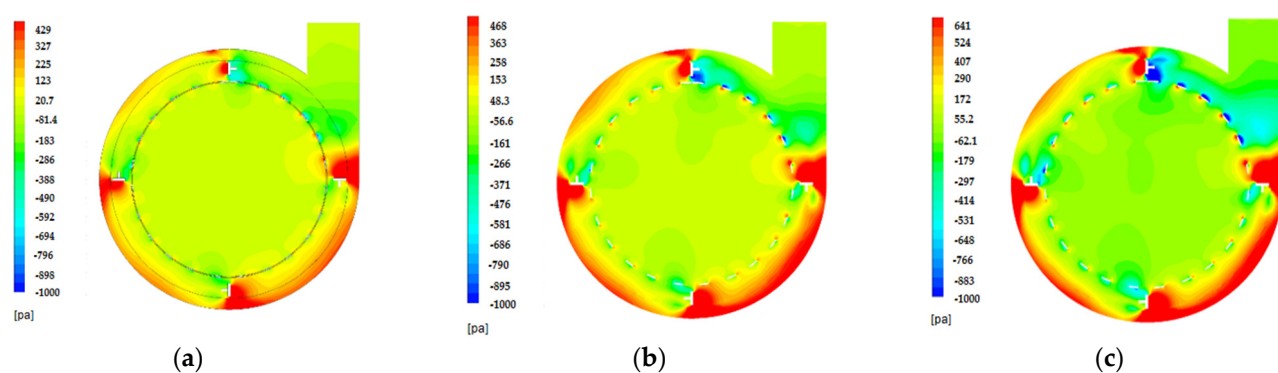

**Figure 10.** Pressure nephogram of section S2 at different rotating speeds. (**a**) Pressure nephogram at 2000 r/min. (**b**) Pressure nephogram at 2500 r/min. (**c**) Pressure nephogram at 3000 r/min.

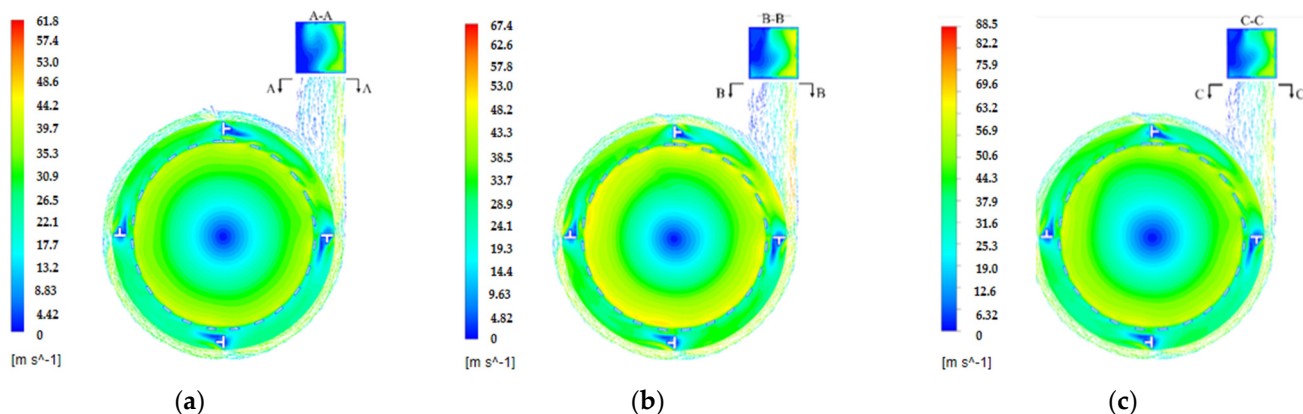

**Figure 11.** Velocity vector nephogram of section S2 at different rotating speeds. (**a**) Velocity vector nephogram at 2000 r/min. (**b**) Velocity vector nephogram at 2500 r/min. (**c**) Velocity vector nephogram at 3000 r/min.

Based on the analysis of the airflow characteristics of the three models in the crushing chamber, although the hammer rack and blades could complete the shear and crushing function in the crushing chamber, due to the low speed of free flow in the non-shear area, it is difficult to timely deliver the crushed materials to the outlet, on the contrary, the materials would easily accumulate and block up in the crushing chamber, resulting in a reduction of productivity. Meanwhile, for lack of restriction on the material size from the sieve, the effective contact time between the blades and materials would be reduced, and the work of the hammer rack and blades alone could hardly ensure the quality of crushing. The setting of the sieve could increase the dwelling time of the materials in the crushing area and enhance the crushing quality. However, the sieve wall surface had a significant blocking effect against the airflow field, so that the speed of free flow was greatly reduced in the flow field area between the sieve outer wall and the chamber wall, and the conveying effect of materials was weakened, the materials would easily block after screening. Influenced by the centrifugal force and gravity in the crushing chamber after crushing, the crushed materials would stick to the wall of the crushing chamber. Because the impeller in this area rotated rapidly and could achieve a good throwing effect after screening the materials, moreover, at different rotating speeds, the airflow speed at the outlet was significantly enhanced so that the screened materials could timely be discharged and the production efficiency of the pulverizer could be elevated.

## 6. Test Verification and Analysis

### 6.1. Test Design and Method

In order to verify the reliability of research results, in this test, a square bale corn stalk pulverizer was taken as the object to test the flow field distribution at idling conditions. The test site is shown in Figure 12. In order to reduce external interference, a windless environment with an ambient temperature of 25 °C was selected for the test [24,25]. A contact-type digital tachometer (TEANS TA8146 series) was adopted to measure the rotor speed, with a speed measurement range of 0–9999 r/min and an accuracy of 1 r/min. An SYT-2000V intelligent digital pressure anemometer (manufactured by Xiamen Zongyi Instrument Co., Ltd., Xiamen, China) was adopted in the test. The testing range of the anemometer is 1.00~50.00 m/s, with an accuracy grade of 1.0FS. In the test, the frequency of the frequency converter was adjusted to achieve a rotating speed of 2000 r/min, 2500 r/min, and 3000 r/min for the spindle, and the airflow velocity and flow rate at the outlet plane were measured.

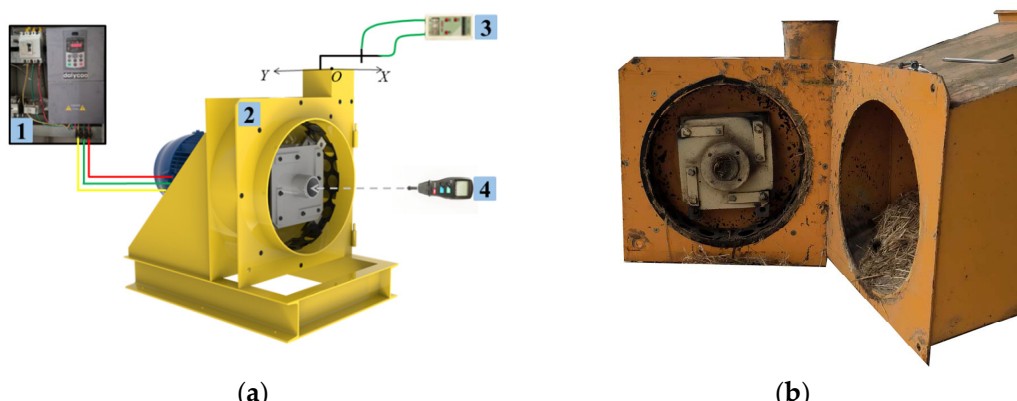

(**a**)                                         (**b**)

**Figure 12.** Testbed for velocity field measurement. (**a**) 1. Frequency converter. 2. Pulverizer. 3. Pressure anemometer. 4. Tachometer. (**b**) Physical map.

In testing the speed at the outlet, based on the coordinate system in SolidWorks modeling, a rectangular cross-section S of the outlet at the point Z = 300 mm, parallel to the XY plane, was selected. The wall near the outlet of the crushing chamber was taken as the X-axis, the wall near the spindle was taken as the Y-axis, and the intersection point of the two wall surfaces was taken as the origin O to establish a new coordinate system. Equal area division was adopted in the test [26]. Divide the rectangular section into nine small rectangles with equal areas, and measure the wind speed at the center of each small rectangle. As shown in Figure 13, three measurement points were taken from the X-axis of Section A, and three measurement points were taken from the Y-axis. Each measurement point was tested three times, and the average values of the test results were recorded.

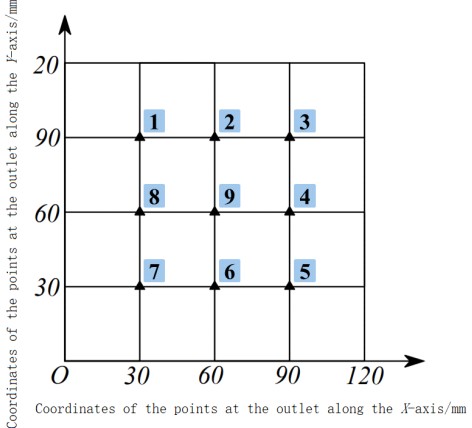

**Figure 13.** Layout of airflow speed measurement points at the outlet.

*6.2. Test Results and Analysis*

Because the results of the numerical simulation were presented in the form of a distribution range, the test results were the airflow speed at a certain point; therefore, in order to more accurately compare and analyze the simulation and test results, the software CFD-POST was applied to extract the data of each testing point among simulation results. The simulation and test results of airflow values of each testing point of the three models at the outlet are shown in Figure 14.

Figure 14(1)–(3) show the airflow nephogram of each model at rotating speeds of 3000 r/min, 2500 r/min, and 2000 r/min at the outlet.

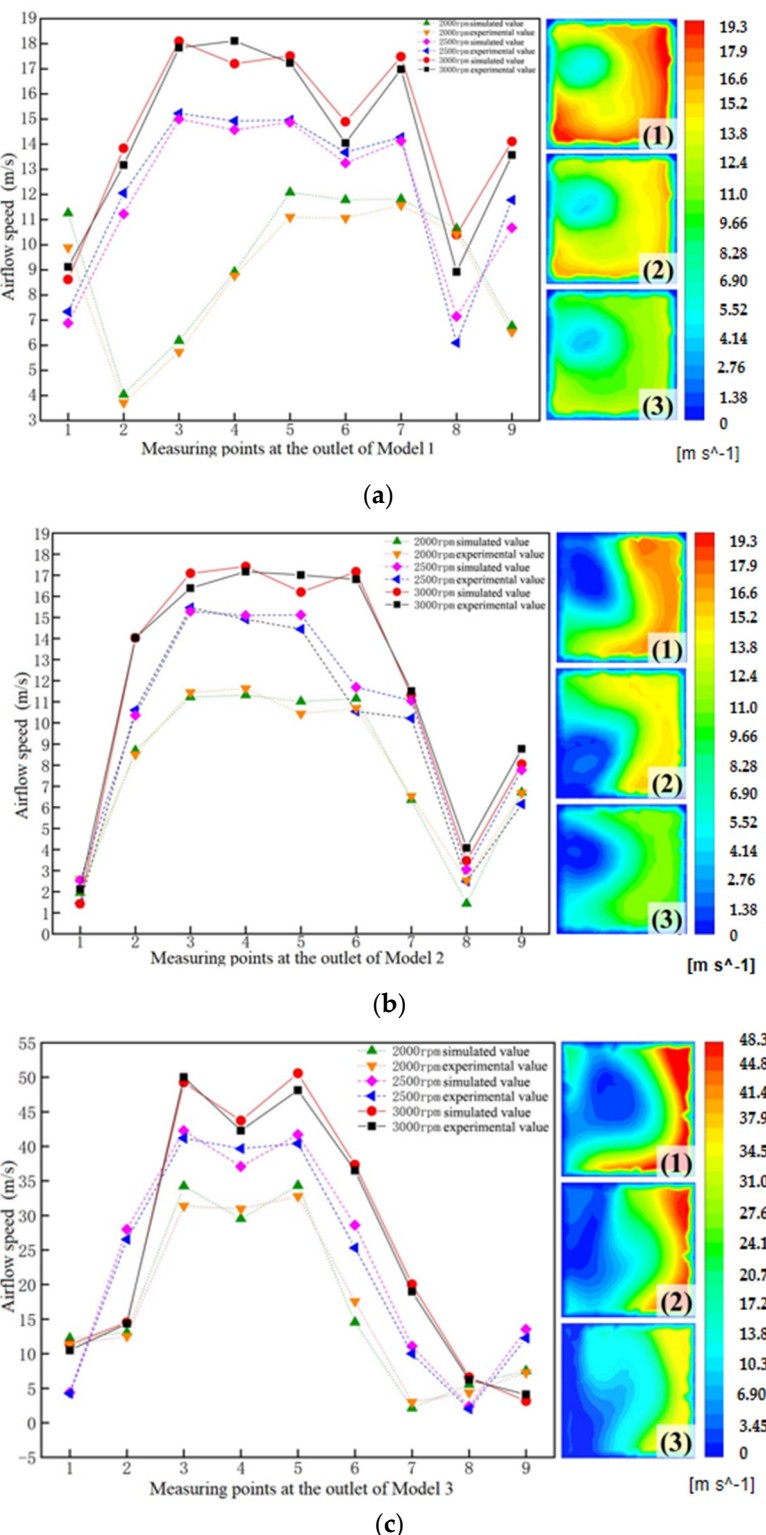

**Figure 14.** Comparison of simulation values and test values of the three models. (**a**) Comparison results of measured airflow speed of Model 1. (**b**) Comparison results of tested airflow speed of Model 2. (**c**) Comparison results of tested airflow speed of Model 3.

It can be obtained from the test results that the airflow speed near the left area of the rotation axis at the outlet was relatively low, while the right side had higher airflow speed. The reason is that the spindle in the crushing chamber rotated anticlockwise, and the hammer blades tilted slightly and disturbed the inner flow field. Then, the high-speed

airflow moves toward the outlet along the wall surface of the crushing chamber. At the same time, the inner flow field of the crushing chamber moves along the tangent line of the throwing blades, and the tangent line is directed to the right area of the outlet, which increases the airflow speed in the right area. The 3rd, 4th, and 5th measuring points were selected from the right area. In the three models, the air speed values show an increasing trend with the increase of rotating speed, and the changes were relatively stable. The maximum air speed values alternate among the 3rd, 4th, and 5th measuring points. At 3000 r/min, the air speed at the 3rd measuring point of Model 1 was 15.23 m/s, the air speed at the 4th measuring point of Model 2 was 17.07 m/s, and the air speed at the 5th measuring point of Model 3 was 49.77 m/s. The measuring points 1, 2, 6, 7, 8, and 9 were selected for the left and middle areas. Due to the significant airflow disturbance at the throwing area and the outlet, unfixed vortices with unfixed vortex centers and radius of rotation were generated in the left area. The values of the six measuring points vary greatly and have no significant change in law.

The trend of changes in simulated and test values was similar. In general, the simulated values were slightly larger. The errors mainly arose from three aspects: insufficient processing, manufacturing, and assembly accuracy of components, resulting in the unqualified sealing of the crushing chamber and errors; the interference generated by the vibration of the working components inside the crushing chamber during high-speed rotation could cause errors in the measured values; in a numerical simulation, the creation of the model was based on simplification and smoothness, ignoring some actual operating conditions and energy loss in the flow field, resulting in higher simulated values.

By analyzing and comparing the simulated and test values at each measuring point, the average error range was between 4.29% and 11.84%, and the overall trend of change was consistent, proving that the numerical simulation on the flow field in the crushing chamber of the square bundle pulverizer was reasonable and could accurately and effectively reflect the distribution of the internal flow field in the crushing chamber of the pulverizer at idling condition.

## 7. Conclusions

By adopting the CFD method to analyze the key components in the crushing chamber, the pressure nephograms and velocity vector of the three models, namely the hammer rack and blades, hammer rack and blades and the sieve, and the sieve and the impeller, at different cross sections at different rotating speeds, and measure the air speed at the outlet of the three models through tests, at last, the following conclusions were drawn after analysis and research:

(1) The hammer rack and blades, the central area of the impeller spindle, as well as the inlet area were negative pressure areas, and the outlet area was the positive pressure area. The airflow speed near the spindle of the pulverizer and near the wall surface of the sieve was low, while the airflow speed near the hammer blades and the throwing blades was relatively high.

(2) The CFD method was used to study the flow field characteristics of three models in the crushing chamber. Based on the simulation results of the three models, it can be concluded that the arrangement of the hammer rack and blades, the sieve, and the impeller in the crushing chamber could effectively improve material crushing quality and conveying efficiency. Moreover, the increase in rotating speed had a positive impact on the improvement of the working efficiency of the entire machine. At a rotating speed of 3000 r/min, the crushing efficiency and conveying efficiency were the highest.

(3) The verification of the simulation results shows that the maximum average error between the simulated and test values was 11.84%, showing a consistent overall trend of change, with the error within a reasonable range.

**Author Contributions:** Conceptualization, J.Z., R.G., X.T. and B.F.; methodology, J.Z., R.G., X.T. and B.F.; software, J.Z. and R.G.; validation, C.Z. and X.Y.; formal analysis, C.Z., X.Y. and S.A.; investigation, J.Z. and R.G.; resources, R.G. and J.Z.; data curation, J.Z., R.G. and C.Z.; writing—original draft preparation, J.Z. and R.G.; writing—review and editing, J.Z., X.T., R.G. and S.A.; project administration, B.F. and J.Z.; funding acquisition, B.F., X.T. and J.Z. All authors have read and agreed to the published version of the manuscript.

**Funding:** This study was supported by the Autonomous Region's Key Research and Development Program project topic: research on straw-based forage utilization equipment technology (2022B02042-3) and the Xinjiang agricultural machinery research and development, manufacturing and application integration project. Research and development, manufacturing, and popularization of precision feeding machinery and equipment for cattle and sheep farms in Xinjiang (YTHSD2022-19) and the promotion of a technical system for the selection of high-efficiency meat goat breeds in the agricultural area of the autonomous region are the focus of this expert task book (xjnqry-g-2306).

**Institutional Review Board Statement:** Not applicable.

**Data Availability Statement:** All data are presented in this article in the form of figures and tables.

**Acknowledgments:** All subject programs are thanked for funding this study.

**Conflicts of Interest:** The authors declare no conflicts of interest.

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
