# Peer review of "CFD-Based Study on the Airflow Field in the Crushing Chamber of 9FF Square Bale Corn Stalk Pulverizer"

_agriculture, doi:10.3390/agriculture14020219_

Round 1

Reviewer 1 Report

Comments and Suggestions for Authors

This paper has a certain value of bale corn stalk pulverizer design, and the academic value is beyond doubt. However, there are some problems with this paper, the suggestions for improvement are as follows.

1. There is a difference in expression for the three simplified 3D models, check and modify.  

2. How to optimize the structure of the 9FF square bale corn stalk pulverizer?  I didn't find it in the paper.

3. What are the factors affecting crushing efficiency?

Comments on the Quality of English Language

Minor editing of English language required. 

Author Response

Dear reviewer and editor:

First of all, thank you very much for taking the time out of your busy schedule to read and modify my article. Thank you for your valuable suggestions. You have corrected all aspects of the structure, content, research methods, and results of my paper, which will play a very important role in improving the quality of my paper.

I have carefully read the comments of the reviewer and carefully revised the paper item by item according to the suggestions. Please check the attachment for details.

Thank you very much for your consideration.

Reviewer 2 Report

Comments and Suggestions for Authors

1.  Section2.1 suggests adding a three-dimensional modeling diagram of the whole machine for easy understanding.

2.  At the end of the first paragraph of Section 2.2, the punctuation is wrong.

3.  Table 1 mentioned in the second paragraph of Section 2.2 does not correspond to Table 1 in the text.

4.  English writing is general.

5.  The third chapter introduces the relevant theoretical methods of fluid mechanics. These recognized principles and methods do not need to be described in a large space in the article.

6.  On the sixth page, the second paragraph assumes that the movement of the hammer relative to the hammer frame lag 5 °, should not assume that should be observed from the actual work ( such as the existing powder machine working situation ) to get or reference to others.

7.  Figure 4 (a) has spelling errors.

8.  The two formulas that appear in Section5.2 are meaningless and need not be specifically listed.

9.  Page 12. The order of â‘ â‘¡â‘¢ is confused.

10.  The sixth chapter is to verify the test proposal to put the physical map.

11.  The annotation of Figure 14 (b) (c) is wrong.

12.  The article analyzes the outlet speed, and there is no analysis on the crushing quality and crushing efficiency. Can these two indicators be quantitatively analyzed?

Comments on the Quality of English Language

Moderate editing of English language required.

Author Response

Dear reviewer and editor:

First of all, thank you very much for taking the time out of your busy schedule to read and modify my article. Thank you for your valuable suggestions. You have corrected all aspects of the structure, content, research methods, and results of my paper, which will play a very important role in improving the quality of my paper.

I have carefully read the comments of the reviewer and carefully revised the paper item by item according to the suggestions. Please check the attachment for details.

Thank you very much for your consideration.

Best regards!

Yours sincerely,

Corresponding author: Bin Feng, Rui Guo

E-mail: fbxjnky@stu.nwupl.edu.cn; goodrui2023@163.com

Author: Jie Zhang

E-mail: 13999880230@163.com

Reviewer 3 Report

Comments and Suggestions for Authors

Abstract:

·       The during the steady – To be written as During the steady

·       CFD – Full form to be written when it is used for the first time

·       Brief quantification of results is required in the abstract

Introduction:

·       Last few sentences are same as that mentioned in the abstract. There should not be repeatability.

2.1 Working principle

·       How 15° is selected? Do the authors have literature support? If this angle is varied, what is the significance on end result?

2.2 Analysis on pneumatic conveying

·       Materials were subject to – To be written as materials were subjected to – This is to be corrected twice

·       Last sentence of the first para – Looks like incomplete sentence

·       Eq (1) & (2): Unit of area is m2 and not m3

3.1 Basic governing equation in the flow field simulation analysis

·       Why incompressible is assumed? Is there any literature available to support this?

·       Some notations, such as ρ, are abbreviated multiple times. Instead a nomenclature table may be added to define all the symbols at once.

·       Eq(6): Previously, ρ was density and now it is shown as pressure. Replace it with ‘p’

4.1: Establishment of the flow field regions based on Design modeler

·       Usual work processes like Naming the file, saving the file should not appear in the technical article. These are internal matters. Remove such sentences.

4.2 Mesh generation based on Mesh: What is the meaning of this heading?

·       Inconsistency in writing Figure as Fig.5 or Figure 5. Maintain uniformity.

·       Table 1: What is the meaning of Total units?

·       Fig. 5: What results Fig 5 convey?

5.1: Setting of boundary conditions

·       Write Moving Reference Frame (MRF). Provide abbreviation.

5.2: Turbulence model in the flow field analysis

·       K and epsilon equations are not numbered

·       Any CFD analysis should have grid sensitivity analysis. It is a must to show in the article.

·       CFD results must be validated either through experiments or analytically. Without validation, CFD results alone can not be believed. This must be included.

5.3 Results and Analysis

·       Why paragraphs are numbered in this section? It is felt that such numbering is not required.

6.1 Test design and method

·       A validation curve must be shown

Author Response

(The authors gave the same response as above.)

Round 2

Reviewer 2 Report

Comments and Suggestions for Authors

Minor editing of English language required.

Comments on the Quality of English Language

1.The first letter of the Fig. 1(a) should be capitalized.

2.Fig. 1(b) is suggested to be drawn according to national standards.

3.The figures and tables in the text should be standardized.

Author Response

Dear reviewer and editor:

First of all, thank you very much for taking the time out of your busy schedule to read and modify my article. Thank you for your valuable suggestions. You have corrected all aspects of the structure, content, research methods, and results of my paper, which will play a very important role in improving the quality of my paper.

I have carefully read the comments of the reviewer and carefully revised the paper item by item according to the suggestions:

Review 2

  1. The first letter of the Fig. 1 (a) should be capitalized.

I have read it carefully and corrected it.

  1. 1 (b) is suggested to be drawn according to national standards.

I have read it carefully and corrected it.

  1. The figures and tables in the text should be standardized.

I read the full text carefully and standardized the pictures and tables.

We apologize for the unnecessary trouble caused by our negligence. We have enabled the revision mode to revise the manuscript again, and have uploaded the revised and clean version of the manuscript respectively.

Finally, I would like to express my thanks again for your guidance, and thank you for reviewing and correcting my revised paper again. I hope we can complete an excellent paper with your guidance and help. I sincerely hope that my paper can be published in your journal.

Thank you very much for your consideration.

Best regards!

Yours sincerely,

Corresponding author: Bin Feng, Rui Guo

E-mail: fbxjnky@stu.nwupl.edu.cn; goodrui2023@163.com

Author: Jie Zhang

E-mail: 13999880230@163.com

Reviewer 3 Report

Comments and Suggestions for Authors

Authors have incorporated all my comments. Paper may be accepted for publication

Author Response

Dear reviewer and editor:

I would like to express my thanks again for your guidance, and thank you for reviewing and correcting my revised paper again. I hope we can complete an excellent paper with your guidance and help. I sincerely hope that my paper can be published in your journal.

Thank you very much for your consideration.

Best regards!

Yours sincerely,

Corresponding author: Bin Feng, Rui Guo

E-mail: fbxjnky@stu.nwupl.edu.cn; goodrui2023@163.com

Author: Jie Zhang

E-mail: 13999880230@163.com